# Cascading hazards of a major Bengal basin earthquake and abrupt avulsion of the Ganges River

Elizabeth L. Chamberlain [1,2,3] ✉, Steven L. Goodbred [2] ✉, Michael S. Steckler[3], Jakob Wallinga [1], Tony Reimann[4], Syed Humayun Akhter[5,6], Rachel Bain[2], Golam Muktadir [7], Abdullah Al Nahian[5], F. M. Arifur Rahman[5], Mahfuzur Rahman[5,8], Leonardo Seeber[3] & Christoph von Hagke[9]

Earthquakes present severe hazards for people and economies and can be primary drivers of landscape change yet their impact to river-channel networks remains poorly known. Here we show evidence for an abrupt earthquake-triggered avulsion of the Ganges River at ~2.5 ka leading to relocation of the mainstem channel belt in the Bengal delta. This is recorded in freshly discovered sedimentary archives of an immense relict channel and a paleo-earthquake of sufficient magnitude to cause major liquefaction and generate large, decimeter-scale sand dikes >180 km from the nearest seismogenic source region. Precise luminescence ages of channel sand, channel fill, and breached and partially liquefied floodplain deposits support coeval timing of the avulsion and earthquake. Evidence for reorganization of the river-channel network in the world's largest delta broadens the risk posed by seismic events in the region and their recognition as geomorphic agents in this and other tectonically active lowlands. The recurrence of comparable earthquake-triggered ground liquefaction and a channel avulsion would be catastrophic for any of the heavily populated, large river basins and deltas along the Himalayan arc (e.g., Indus, Ganges, Brahmaputra, Ayeyarwady). The compounding effects of climate change and human impacts heighten and extend the vulnerability of many lowlands worldwide to such cascading hazards.

Avulsion is a fundamental delta-building process that sustains coastal landscapes by redistributing the focus of sediment deposition over time to offset basin subsidence, sea-level rise, and localized aggradation of channel belts. The processes that govern avulsion remain debated[1-4], and the dominant mechanisms may vary between relatively high-gradient, coarse-sediment fan settings and relatively low-gradient, fine-grain coastal settings as well by climate and sediment supply[5]. In its simplest form, avulsion results when channel belts become superelevated relative to their floodplain and shift to a new course through gradual abandonment by stream capture[6,7]. This mechanism is largely supported by autogenic river processes, but allogenic forces such as tectonic deformation or earthquakes may

[1]Soil Geography & Landscape group and Netherlands Centre for Luminescence dating, Wageningen University, Wageningen, The Netherlands. [2]Department of Earth & Environmental Sciences, Vanderbilt University, Nashville, TN, USA. [3]Marine & Polar Geophysics, Lamont-Doherty Earth Observatory, Columbia University, Palisades, NY, USA. [4]Mathematics & Natural Sciences, University of Cologne, Cologne, Germany. [5]Department of Geology, University of Dhaka, Dhaka, Bangladesh. [6]Bangladesh Open University, Gazipur, Bangladesh. [7]Department of Environmental Science, Bangladesh University of Professionals, Dhaka, Bangladesh. [8]Department of Oceanography, Noakhali Science and Technology University, Noakhali, Bangladesh. [9]Department of Environment & Biodiversity, Geology and Physical Geography, University of Salzburg, Salzburg, Austria. ✉e-mail: liz.chamberlain@wur.nl; steven.goodbred@vanderbilt.edu

contribute to the avulsion of rivers in seismically active basins[8,9]. In regions such as the Himalaya, ongoing collision, uplift, and erosion have produced thick sedimentary sequences in both foreland and continental-margin basins that lie adjacent to the active seismogenic zones[10]. Among the largest and most heavily populated of these basins are the major rivers valleys and their coastal delta systems, including the Indus, Ganges, Brahmaputra, and Ayeyarwady.

The seismic risk to such lowlands is broadly recognized[11–16] but remains poorly constrained with respect to response magnitude and cascading hazards[17]. In part, the large size of these basins means that they often extend >100 km from the nearest faults and may only be impacted by larger, less frequent events for which there are scant historical records. Instrumental and historical timescales of years to centuries are likely to miss large, rare events with millennial recurrence intervals[18]. For example, a recent satellite-based synthesis of channel avulsions found no statistical correlation with earthquakes occurring in that instrumental timescale[5]. However, millennial-timescale reconstructions suggest evidence of earthquake impacts on inland river channel geometries in the New Madrid seismic zone, USA[19,20], a region which corresponds to channel-belt avulsion sites of the Mississippi River over the Holocene[21,22]. Similarly, earthquakes in India's Rann of Kutch region have caused widespread liquefaction and flooding from fault-induced water diversions[16,23–25]. Paleoenvironmental reconstructions alongside archaeological sources indicate this may be a recurring event[16,26]. The displacement of river channels proximal to ruptured faults is previously recognized[9] yet the impact of seismicity on more distal lowland channels is not known or well documented. Although often far from seismogenic sources, the potential for earthquake impacts to lowland river-channel networks is amplified by the propensity of mud-capped, saturated fine-medium sands to liquefy, the greater relative effect of land-surface deformation and water routing on low-gradient landscapes, and fracture or failure of riverbank levees through lateral spreading[27,28].

As a vivid and empirical example of seismic risk to lowland basins, we present evidence from the sedimentary archive of the Bengal basin (Fig. 1) showing abrupt and complete avulsion of the mainstem Ganges River coincident with a major paleo-earthquake. The river-channel avulsion is evidenced by the cessation of sand transport and overbank mud deposition at ~2.5 ka along an 85-km reach of this Ganges paleo-channel belt. Located within 1-km of the abandoned channel are large

(30–40-cm wide) sand dikes which breach the entire ~2.6–2.5 ka floodplain sequence. Such an archive is previously undocumented in the Bengal basin. The characteristics of the sand dikes, including their orientation, fracture bifurcations, brecciated sediment clasts, and liquefied sedimentary structures, all rule out a non-seismic origin and indicate that the dikes were formed by sediment liquefaction during an earthquake. A suite of thirteen precise optically stimulated luminescence (OSL) ages obtained from quartz silt[29] constrain the apparent coeval occurrence of the avulsion and seismic events. These findings support a largely unrecognized allogenic avulsion mechanism and highlight the elevated risk of cascading earthquake hazards in river deltas and other lowland basin settings.

## Results

### Paleochannel archive

The 150,000 km² Bengal basin is part of the trailing-edge continental margin of the eastern Indian subcontinent. Its northern and eastern perimeters lie at convergent plate boundaries that actively deform the kilometers of Neogene and older fluvio-deltaic sediments that infill the basin[30]. The Holocene delta sequence is up to 90 m thick and its sediments are primarily supplied by two of Earth's largest rivers – the Ganges and Brahmaputra[31].

In the central delta plain, ~45 km south of the modern Ganges River, the scar of a 1.0–1.7 km-wide meandering paleochannel is evident in the delta's surface morphology (Fig. 1a, b). The relative elevation of the present-day paleochannel surface is ~3 m below the adjacent levee and floodplain surfaces; the paleochannel is consequently poorly drained and used for rice cultivation (Fig. 2). The wide paleochannel is partially infilled with muds that are 3–15 m thick and abruptly overlie a well-preserved sandy channel bed. This sand surface defines the meandering paleochannel's point-bar front and thalweg (Fig. 2a). OSL ages of the upper point-bar and channel-fill sands both place the last occurrence of active sand transport at 2.61 ± 0.11 ka with an abrupt transition to mud infilling at 2.48 ± 0.14 ka (Fig. 2a). Ages from the adjacent floodplain show coeval, rapid overbank aggradation producing a ~ 4 m-thick sequence dating from 2.63 ± 0.15 ka at its base to 2.49 ± 0.10 ka near the modern land surface (Fig. 3a). Combined with the paleochannel results, these data indicate that a large, aggrading river system was active here around 2.6 ka, and the cessation of sand transport and overbank deposition indicate abrupt

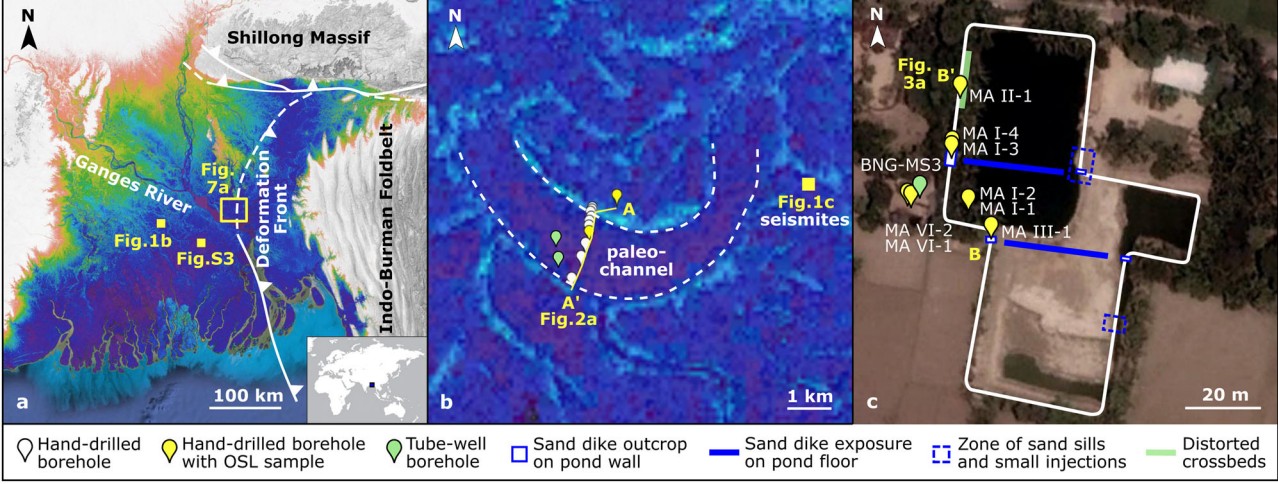

**Fig. 1 | Abandoned channel scar and seismite locations within the fluvial and tectonic setting of the Bengal basin. a** The central Bengal basin is shaped by the Ganges River, the Indo-Burman megathrust deformation front and foldbelt growth, and the Shillong Massif thrust and uplift, among its fluvial and tectonic agents. **b** Coring and luminescence sampling was executed across a vast, underfilled paleochannel evident on the Ganges floodplain surface and at an adjacent dry pond with seismite features. **c** The perimeter of the drained pond at the time of fieldwork. The pond walls and floor revealed extensive sand dikes, fluidized muds, and liquefied sand layers, which were sampled for luminescence dating. The background images for panels a and b show relative surface topography from NASA SRTM elevation data.

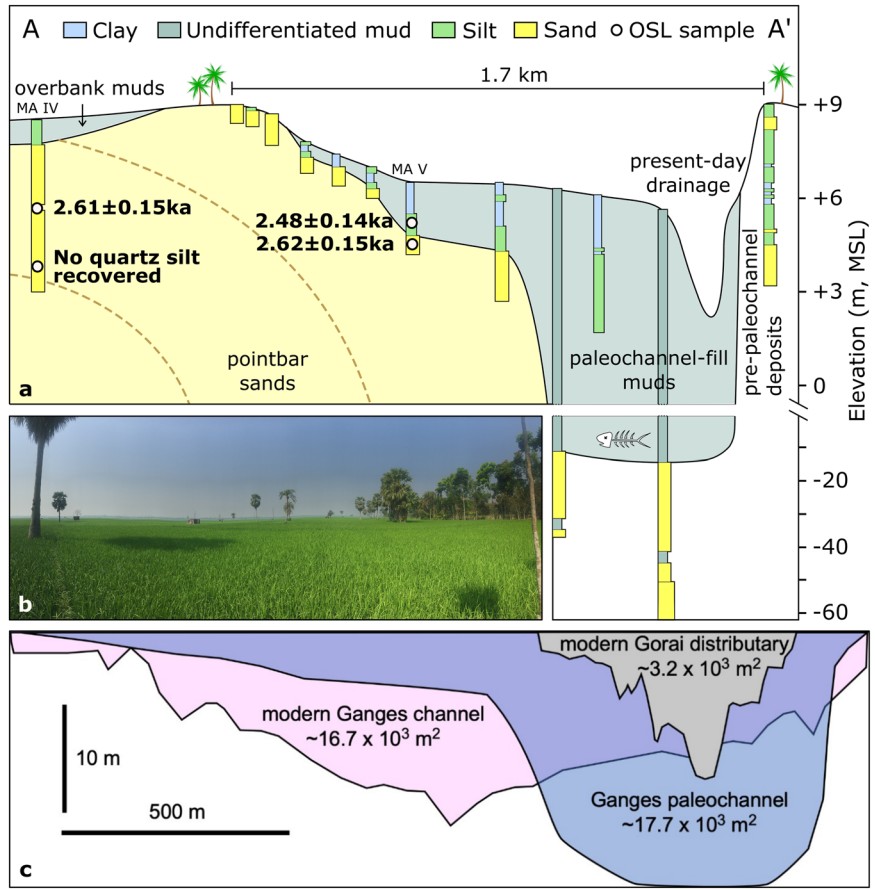

**Fig. 2 | Stratigraphy and chronology of the abandoned paleochannel. a** Hand and tube-well coring of the paleochannel revealed a typical meandering-channel bed profile with a thalweg depth of ~20 m below the surface. The channel was only partially infilled with fine muds, atypical for abandoned channels of the mainstem Ganges River. Channel-sand ages yield a weighted mean of 2.61 ± 0.11 ka, and the infilling muds luminescence dated to 2.48 ± 0.14 ka. **b** View from the pointbar of the paleochannel, looking across to the outer bank, shows the scale of the feature.

**c** Comparison of cross-sectional areas of the modern mainstem Ganges River at Hardinge Bridge, the modern Gorai distributary (a secondary offtake), and this paleochannel supports interpretation of this feature as a former mainstem Ganges channel. Note that the different morphology for the modern and paleo cross-sectional profiles simply reflect their differing locations along straight and meandering channel reaches, respectively.

channel abandonment around 2.5 ka. Prior to abandonment, the average overbank sedimentation rate was a rapid 1.9 ± 0.8 cm/yr, which is consistent with rates from other proximal floodplain settings in the delta[32,33]. The abrupt and widespread cessation of sand transport along the entire paleochannel belt is supported by a coeval age of 2.62 ± 0.16 ka for the shallow-buried sands of a preserved mid-channel bar top ~85 km downstream (Fig. S3). These shallow, contemporaneously deposited bar-top units lie along a low-elevation swath (Fig. 1a, S3) indicating the continuous absence of fluvial clastic deposition since the river course was abandoned at ~2.5 ka.

The sands at these upstream (MA IV-1, MA IV-2, MA V-1) and downstream (RD III-1 and RD III-2) locations consistently yield bulk-strontium values ranging 83 ± 3 to 98 ± 3 ppm, which unambiguously fall within the range for Ganges River provenance for the central and western delta[34] (Supplementary Methods 3. Geochemical and grain-size measurements). Furthermore, the scale of the paleochannel indicates that it represents the mainstem pathway of the Ganges River and was not a distributary. The paleochannel is 1.0–1.7 km wide with a cross-sectional area of ~17.7 × 10³ m² based on the depth to sand from our borehole transect (Fig. 2). This cross-sectional area is within 5% of the ~16.7 × 10³ m² value for the modern Ganges channel at Hardinge Bridge, where the river is 1.5 km wide (Fig. 2c)[35]. In contrast, modern distributary channels of the Ganges River, including the Gorai and Bhagirath-Hooghly distributaries, are only 0.2–0.5 km wide in most places and rarely exceed 0.6 km. These distributary channels have

cross-sectional areas of 1–2 × 10³ m², which are only ~10% that of the modern mainstem river and the abandoned paleochannel (Fig. 2c).

## Sand-dike archive

One kilometer east of the abandoned channel (Fig. 1), two large, linear sand dikes and their subordinate fractures rupture the thick, muddy floodplain sequence and reflect major liquefaction of underlying sand units (Figs. 1, 3). Terrestrial clastic dikes such as these are commonly associated with earthquakes, wherein intensive shaking can fluidize buried saturated sediments and generate sufficient pore-fluid pressure to drive injection of confined sediments into overlying deposits. Modern and ancient examples of earthquake-triggered liquefaction and sand-dike emplacement can be found worldwide[28,36–39]. Clastic dikes may also form through non-seismic mechanisms such as riverbank slumping, load-induced pressurization, and wind events[40,41]. The dikes we identified consist of moderately well-sorted, fine sands (d50 = 200–250 μm) that breach a nearly 4-m thick capping unit of moderately sorted silty muds (d50 = 35–60 μm). Both units are typical of river-channel sands and floodplain deposits in the central Bengal delta, respectively[42]. The size and sorting distribution of the sands fall squarely within the ideal range for liquefaction[41,43,44], as do most of the shallow (<30 m depth) sands across the lower Bengal delta (Fig. 4) and elsewhere in Bangladesh[45].

The two main dikes are ~15 m apart and lie parallel to one another at a roughly east-west strike (Figs. 1c, 3). The dikes are both 10–15 cm

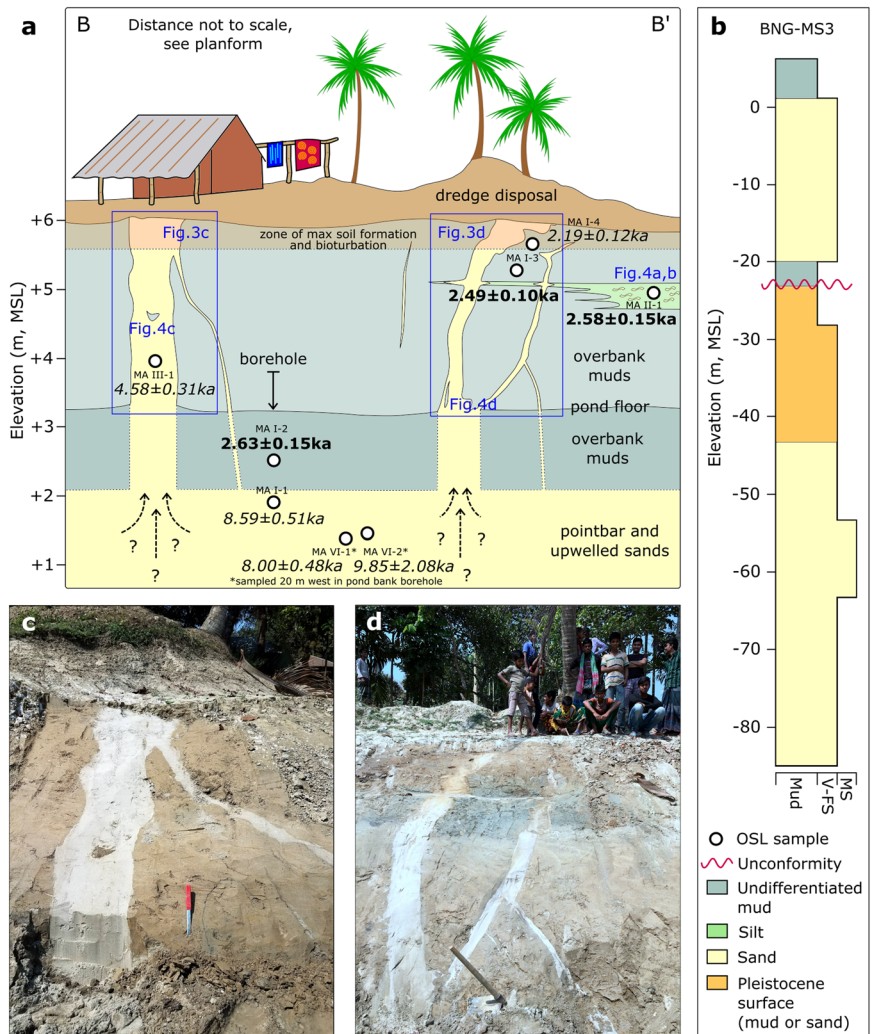

**Fig. 3 | Lithostratigraphy and chronology of sand dikes. a** Depiction of shallow lithostratigraphy, features, and location and results of OSL dating of sand dike features and intruded sediment. The OSL ages are stratigraphically consistent and constrain overbank deposition to ~2.6–2.5 ka, contemporaneous with the activity of the adjacent river channel (Fig. 2a). **b** Deep lithostratigraphy obtained from tube-well drilling shows a mud-confined sand at ~0–20 m elevation overlying Pleistocene-aged sands. **c, d** ~20–40 cm-wide sand dikes bisect the floodplain cap exposed along the pond walls.

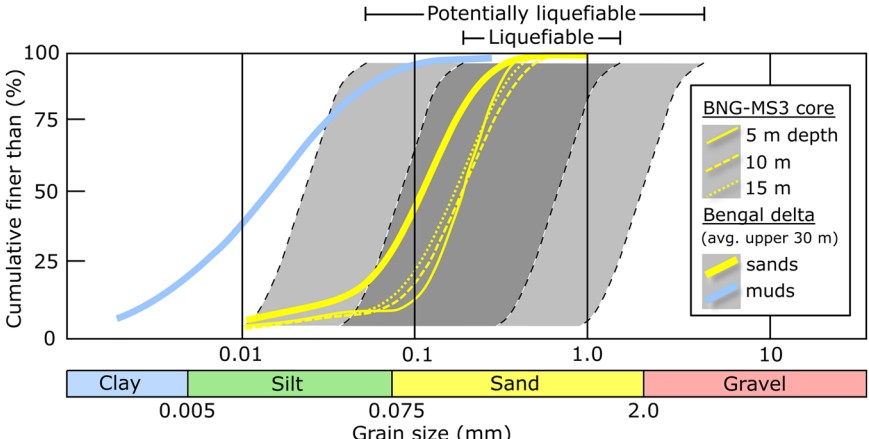

**Fig. 4 | Sediment texture and liquefaction potential.** Grain-size distribution plot showing the shallow sands from borehole BNG-MS3 at the study site (Figs. 1, 3) and the average upper 30-m of Holocene sands and muds of the lower Bengal delta[42]. The range of grain-size distributions susceptible to liquefaction[41,43,44] indicate that sands at the study site and across the lower Bengal basin are highly liquefiable.

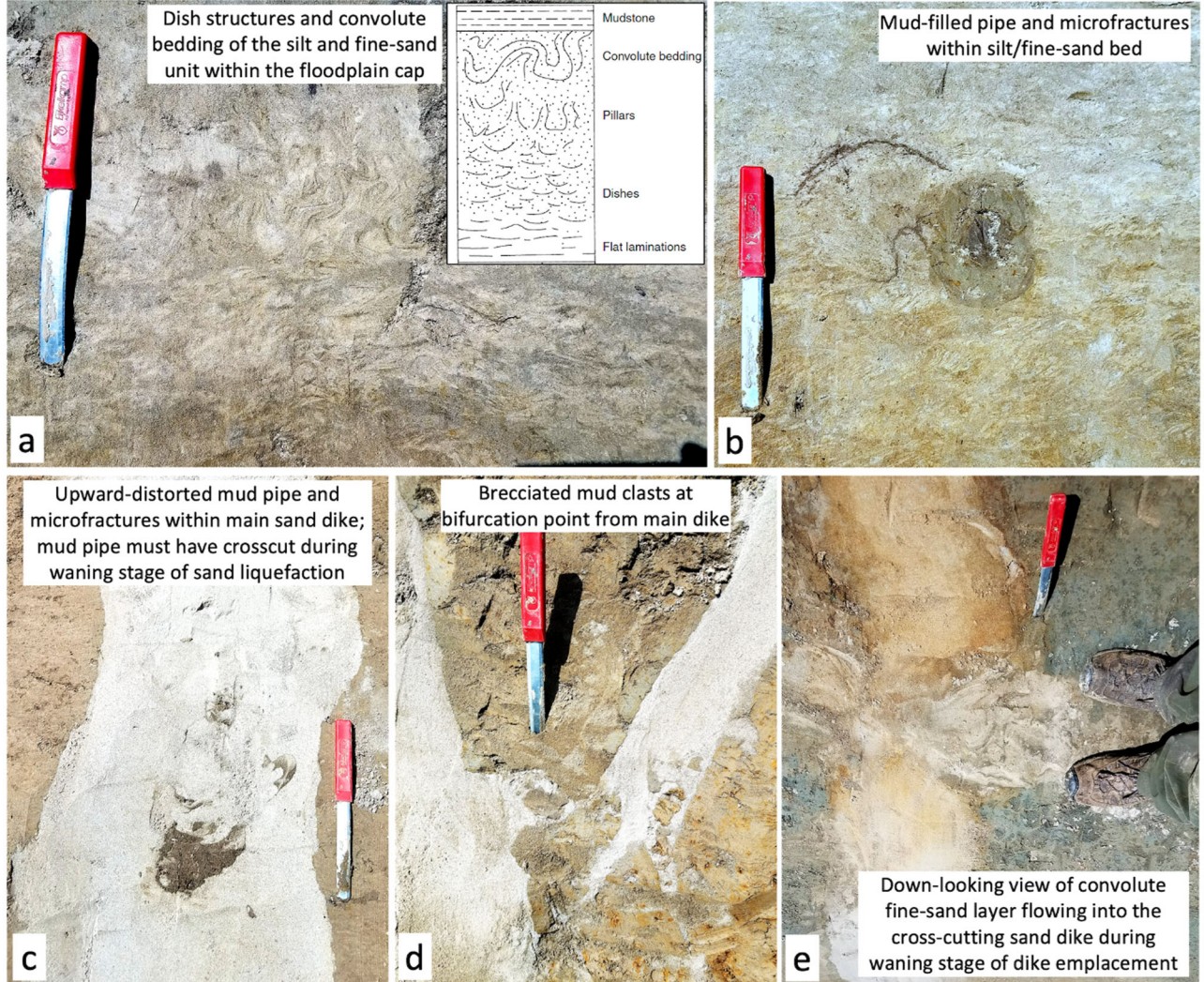

**Fig. 5 | Subordinate seismite features.** Numerous other seismite features were evident in the sediment exposure including (**a**) dish-structured and convolute-bedded sand layer (inset from Obermeier, 2009[111]), (**b**, **c**) mud filled pipes and microfracture, (**d**) brecciated mud clasts, and (**e**) mixing of the convolute sand layer (panel a) and crosscutting sand dike.

wide at their base and widen to 30–40 cm in the upper few meters before reaching ground level (Fig. 3). In this upper zone, the main dikes yield a number of subordinate dikes (5–10 cm wide) that often conjoin one another to form a connected network of structures.

The margins of the sand-dike intrusions are sharply bound and quasi-linear with some angular edges and fractures. Brecciated mud clasts (1–15 cm) are regularly incorporated within the dike sands, but they are most common at bifurcation points of the subordinate dikes (Fig. S1c), where the clasts appear to disrupt flow of the fluidized sands. This arrangement suggests that the subordinate dikes formed in succession after the main dikes and may reflect minutes of sustained ground-surface deformation during emplacement. The large dikes reported here comprise unconsolidated Holocene sands that intruded actively aggrading overbank muds.

Together, the large width (5–40 cm) and complex structure (e.g., bifurcating and intersecting fractures, brecciated clasts) of these Ganges sand dikes suggests sustained, high-energy disturbance that is consistent with a seismic origin. Fracturing and sand injection due to riverbank slumping can be ruled out because of the dikes' orientation normal to the paleochannel (Fig. 1b, c); in contrast, bank extension would cause tensile fractures roughly parallel to the channel[39]. The

timing of sand dike emplacement is well constrained by OSL ages of the capping floodplain muds breached by the dikes. This floodplain deposit yields burial ages of $2.63 \pm 0.15$ ka for muds near its base at 3.5 m depth, as well as an age of $2.58 \pm 0.15$ ka for the silt-to-very-fine-sand layer at ~1.0 m depth and $2.49 \pm 0.10$ ka for muds ~0.8 m below the present-day ground surface (Fig. 3a).

When viewed in cross-section, the scale and architecture of the dikes are similar to those associated with sand blows generated by the 1811–1812 New Madrid earthquakes near the Mississippi River[28,38]. Similar surface expressions of liquefaction (i.e., sand blows) were also reported across the Bengal basin following the 1897 Shillong Massif earthquake[46], but to our knowledge there is no comparable record of vertical exposures like we present here.

**Subordinate seismite archive**

In addition to the main sand dikes, other subordinate deformation features (Fig. 5) also support a seismic origin. The first example is the disturbed bedding of a 30-cm thick unit of silt-to-very-fine sand (<125 μm), which is positioned within the capping mud unit ~1 m below the ground surface (Fig. 5a). The decimeter-scale, silt-sand unit is finely laminated and typical of small splays or overbank deposits in the

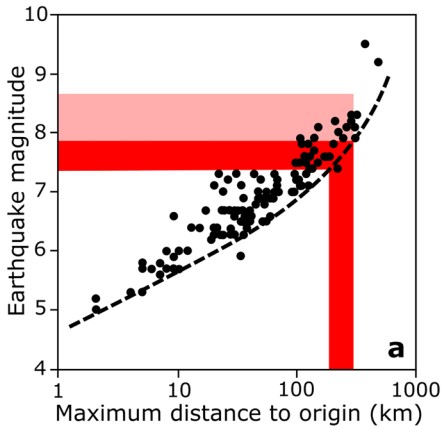
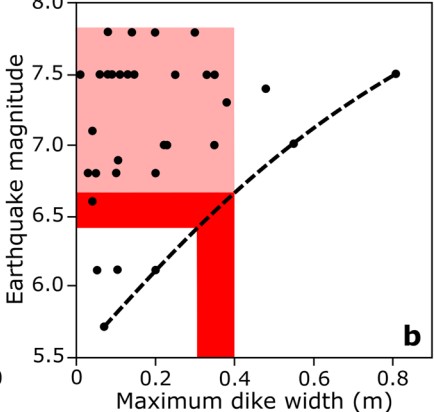

**Fig. 6 | Earthquake magnitude reconstructions.** Reconstructions of earthquake magnitude drawn from global datasets relating to (**a**) distance[39] and (**b**) dike width[37] suggest at least a ~M 6.5 and likely a ~M 7.0–8.0 paleo-event. The red zones indicates the minimum magnitude values based on the range of potential distances to the two likely source areas (Shillong Massif at 200–300 km, and the Indo-Burman subduction zone at 180–280 km) and widths (0.3–0.4 m) of the two primary dikes. The pink zones indicate typical values for the observed ranges.

proximal floodplain, but in this example the laminae are strongly convoluted from being partially fluidized after deposition. Since the fine sands of this fluidized layer also flow into the margins of the sand-dike (Fig. 5e), it is required that both sand bodies were fluidized contemporaneously. Furthermore, fluidization of the silt-sand unit suggests that it was under-consolidated and thus not very old when the sand dike breached it during the earthquake, consistent with the OSL ages showing it to have been deposited just over a century before overbank deposition ceased following the river avulsion (Fig. 3a).

Other subordinate features that help to constrain the timing and mechanisms of disturbance are mud-filled pipes and micro-fractures that crosscut the sand dikes and capping mud unit. These pene-contemporaneous intrusions appear as roughly cylindrical pipes of dark brown mud (3–10 cm diameter, Fig. 5a, b) that all have arcuate, mud-filled micro-fractures radiating from their centers (Fig. S1b). We interpret these dark-colored sediments as disseminated organics and fluidized muds that were generated at the ground surface and infilled void spaces created by the sustained shaking of trees and their root systems. It is the ubiquitous presence of mud-filled microfractures radiating from these mud pipes that indicates the surrounding sediments were being actively compressed by tree-root motion. The possibility that these features were formed during a wind event can be ruled out by the mud pipes and micro-fractures that formed within one of the sand dikes (Fig. 5a); in this instance, the mud pipe and its microfractures are both deformed upward in the direction of sand injection, indicating that the mud pipes and sand dikes formed contemporaneously – or more specifically, the mud pipes probably formed during the waning phase of dike emplacement, enabling their preservation. The abundant formation of mud pipes also suggests that the earthquake occurred during the wet season, when standing surface water and saturated soils make fine surface sediments subject to fluidization.

### Event reconstruction

The preservation of large, extensive, and under-filled channel scars such as those documented here (Fig. 1b, Fig. S3) is rare for the Ganges and Brahmaputra rivers, because their high-water discharge and sediment loads typically infill old channel courses with sand during the avulsion process[47,48]. Thus, (i) the scale and distribution of the abandoned channel reaches, (ii) the abrupt cessation of sand deposition, (iii) the muddy and only partial infilling of the paleochannel, and (iv) its enduring, topographically low surface expression all suggest that this was an abrupt avulsion involving a major reorganization of the Ganges River system. In contrast, a gradual autogenic avulsion[6,7] would be

inconsistent with the sharp termination of both sand transport and overbank deposition along this 85-km channel-belt reach. Furthermore, OSL ages from the cutbank floodplain (Fig. 3) indicate that deposition had been active for only a few hundred years (from ~2.7 ka to ~2.5 ka). This two-century period is considerably shorter than the ~1–2 millennia recurrence interval for autogenic avulsion[49] or delta-lobe cycles[50] in the Bengal delta.

Like the record of channel avulsion, the archive of clastic sand dikes allows for a partial reconstruction of the associated paleo-earthquake event. Extensive research on earthquake impacts has yielded empirical relationships between earthquake magnitude and (a) the distance at which liquefaction occurs and (b) the width of resulting sand dikes[37,39,51] (Fig. 6). The Ganges floodplain sand dikes presented here are located >180 km from the two closest seismically active zones that occur around the Shillong Massif ~200–300 km to the northeast[52] and along the locked portion of the Indo-Burman subduction zone ~180–280 km to the east[12,53] (Fig. 1 and S8). The generation of liquefaction at these distances suggests a minimum earthquake magnitude of $M$ 7.5–8.0 (Fig. 6a), with event scales of $M > 8.0$ more typical for the major liquefaction and sand-dike formation we document[39] (Fig. 6a). Furthermore, the seismogenic zones at Shillong Massif and the Indo-Burman subduction regions are both capable of producing $M > 8$ earthquakes[54]. Shillong Massif was subjected to a ~M 8 earthquake in 1897[46,55,56] that generated widespread liquefaction in Bengal basin[46,57]. The Indo-Burman subduction zone along the Chittagong-Arakan coast produced a ~M 8.5 earthquake in 1762[58,59].

The width of sand dikes can also be used as a general proxy for earthquake magnitude, and these Ganges floodplain dikes are comparable to the largest terrestrial sand dikes reported in a recent global compilation[37] (Fig. 6b). Their 30–40 cm width implies a minimum $M \sim 6.5$ event, but most similarly sized dikes are associated with much stronger $M > 7$ events (Fig. 6b). Taken together, both size and distance relationships suggest that the ~2.5 ka earthquake was likely in the range of $M$ 7.0–8.0. Such an event could originate as a $M$ 7.0 splay-fault rupture in the Indo-Burman ranges 180+ km to the east, or as a larger $M$ 8.0 megathrust rupture produced at either the Shillong or Indo-Burman fronts. Regardless of the earthquake's specific source or magnitude, its impact was immense.

Collectively the sand dikes and abandoned paleochannel record the cascading hazards of far-reaching subsurface liquefaction and an abrupt avulsion of the mainstem Ganges River, each triggered by a major (~ $M$ 7.0–8.0) earthquake that originated >180 km away. The relationship between these cascading hazards is that earthquake-generated ground roll and shallow-sediment liquefaction are plausible

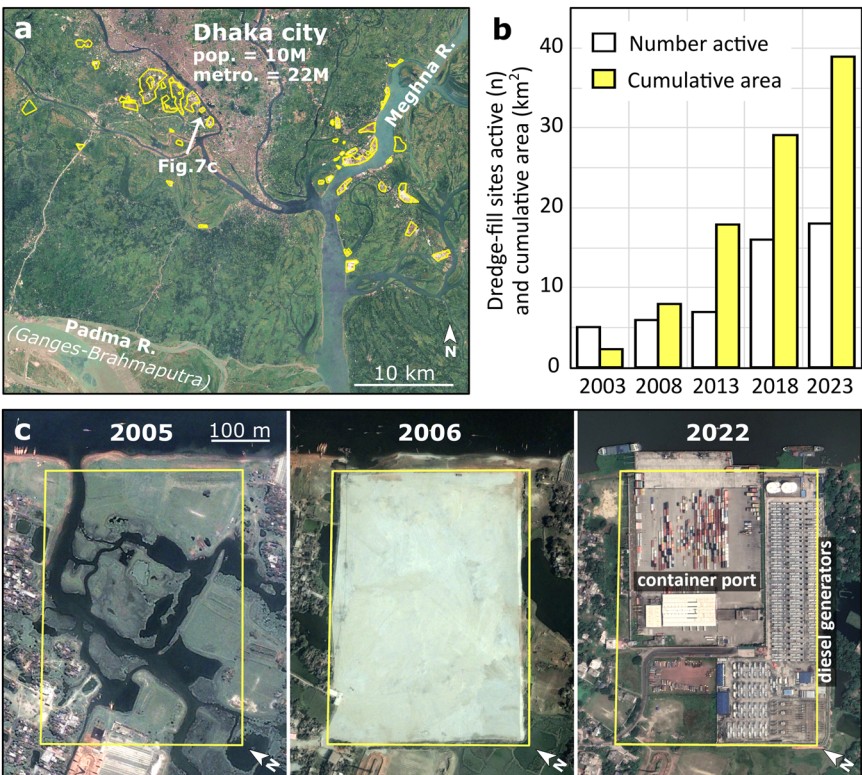

**Fig. 7 | Expansion of dredge-fill construction. a** Dredge-fill construction sites (yellow outline) since 2003 in the vicinity of Dhaka, a capital megacity. Typically 3–4 m of medium-fine-sized river sands are pumped onto the low-lying floodplain to form the construction base. **b** The number and cumulative area of dredge-fill sites on Holocene deltaic lowlands near Dhaka has increased in the past two decades; nearly 30 sites now encompass ~4000 hectares (40 km²) of new development. **c** Time-series example of a 20-hectare dredge-fill site near Dhaka, now serving as a major container port and diesel power station. Images and data in (**a, c**) are from Google Earth Engine[110].

mechanisms for channel avulsion by rerouting surface water on the low-gradient delta plain (~10⁻⁴ to 10⁻⁵ slope), where water flow paths are susceptible to subtle elevation or topographic changes. Recently, differential subsidence hypothetically linked to major Bengal earthquakes in 1762 and possibly ~900 CE has been inferred from the rapid burial and preservation of salt kilns at coastal archaeological sites[60,61] and uplifted micro-atoll corals[62].

Similar observations of rapid topographical change along the banks and proximal tributaries of the Mississippi River are documented for the 1811–1812 New Madrid earthquakes, resulting in what Charles Lyell referred to as "the sunk country"[63,64]. Water-flow disruptions have also occurred in the Rann of Kutch region of western India, where the 1819 Allah Bund ('Dam of God') earthquake generated an 80-km-long fault scarp damming several local rivers[16,25,65]. Prior seismic activity between 712 and 1361 CE is thought to have transformed parts of the Rann of Kutch from a coastal lowland with navigable channels to the seasonal saline lake persisting today, which Lyell referred to as "neither land nor sea"[16,26,66]. Yet, it is important to keep in mind the limitations of historical records and archaeological correlations[29]. The environmental impacts of the New Madrid and Kutch earthquakes are descriptively rich, but the relative contributions of river flooding versus tectonic subsidence are difficult to discern in these older accounts.

If confirmed for the Bengal basin, rapid differential subsidence[60–62] would be consistent with the potential for earthquakes to abruptly alter delta-plain elevations and river gradients. We further infer that the earthquake and avulsion occurred during the wet monsoon season when (i) sediments are saturated and more prone to failure, (ii) liquefaction potential rises with water loading and overpressure, (iii) lateral water gradients increase during bankfull river discharge, and (iv) high water discharge is more capable of

channelizing a new flow path. The 1950 Assam earthquake occurred in August during the monsoon, triggering extensive landsliding and bank failures, with the vastly increased sediment load creating changes in elevation, width, and braiding along the Brahmaputra River[67,68]. Water loading may also be a plausible trigger for major fault rupture, as the mass of monsoon-fed water that seasonally inundates the Bengal basin is sufficiently immense to cause up to 60 mm of elastic strain in the upper lithosphere[69].

## Discussion

Bankfull, high-discharge flood events are common annual occurrences in the monsoon climates of southeast Asia[70–72], where most large river basins are also tectonically active. Based on the results presented here, these populous regions may be susceptible to added risk from the cascading hazard of earthquake-triggered river-channel avulsions. Monsoon precipitation has varied over past millennia and is forecast to yield both increased precipitation and sediment load due to anthropogenic climate change[73–75]. Impacts of climate change are not restricted to the Himalaya, and many lowland rivers worldwide have experienced unprecedented high-magnitude flood events in recent decades related to snow melt[76], increasing frequency and magnitude of tropical storms[77,78], and extreme rainfall events in their catchments[79]. In addition to directly increasing stream power and destabilizing banks, enhanced flooding can drive transient lithospheric compression through water loading. Like the elastic strain observed in the seasonally flooded Bengal basin (up 60 mm)[69], record-breaking rainfall during Hurricane Harvey on the Texas coast, USA in 2017 caused up to 21 mm of elastic compression of Earth's crust[80]. While flood-risk forecasts are somewhat geographically variable, there is broad consensus that flood risk is increasing as a direct consequence

of a globally warming climate[81–84]. Considering high-magnitude bankfull discharge as a setup for an earthquake-triggered avulsion, climate change may increase susceptibility of these basins to unanticipated landscape responses as documented here.

Humans also play a direct role in landscape and societal vulnerability to cascading hazards through development practices and water extraction, storage, and diversion. For instance, engineering solutions that constrain waterways increase flood risk by lowering a river system's capacity to accommodate the high-magnitude events that are becoming more common, often with catastrophic results[85–87]. A prominent example is the 2008 avulsion of the Kosi River, a large tributary of the Ganges River, which shifted course by over 120 km during a seasonal flood due to the compounding effects of downstream river control structures[88,89]. Moreover, urbanization, industrialization, and population pressure extend the potential damages of flooding, liquefaction, and avulsion to lowland cities and agricultural plains[81,90,91]. In sedimentary basins such as the Bengal and others draining the Himalayan arc, growing urban centers and industrial development are commonly constructed on saturated, under-consolidated Holocene sands susceptible to liquefaction[92–94]. This is acutely illustrated by dredge-fill construction sites around Dhaka (Fig. 7a, Fig. S10), a rapidly growing megacity and the capital of Bangladesh, where much new construction has been forced onto a flood-prone deltaplain that is commonly raised above water levels by adding ~3–4 m of dredged river sand. Now covering almost 40 km² of reclaimed bottomland in the last two decades (Fig. 7b), these highly liquefaction-prone dredge-fill sites (Fig. 4) support central private or public industrial infrastructure such as transportation hubs and power stations (Fig. 7c, Fig. S11).

Prior research on cascading hazards in coastal settings has recognized tsunami flooding and landslides due to earthquakes and emphasized the need for better planning and preparation for such cascading events[95,96]. We show that earthquake-driven avulsion of large river networks is another real, yet largely unrecognized, threat to seismically active lowlands. An earthquake and resulting channel avulsion could together cause widespread sediment liquefaction and extensive river flooding, in addition to the direct damage of seismic shaking to infrastructure. Such cascading hazards of a major earthquake, widespread liquefaction, and river-channel avulsion would be a devastating modern occurrence in the Bengal basin and a risk that warrants further study. Numerous sedimentary basins in tectonically active settings worldwide such as those of the Ayeyarwaddy, Bisagno, Chao Phraya, Colorado, Copper, Fraser, Indus, Jordan, Klamath, San Joaquin, Santa Clara, Yangtze, and Yellow rivers occupy similar situations[79,97–106] and may face unrecognized hazards as the result of seismic risk.

## Methods

### Sedimentology and stratigraphy
Stratigraphy was determined up to 5.8 m depth by hand coring with an Edelman hand auger and gouge for 2 boreholes (one including the pit exposure) at the seismite site, 11 boreholes in and on the banks of the paleochannel, and one borehole in the floodplain ~85 km southeast toward the Bay of Bengal. These shallow strata were described in 10-cm intervals with attention to sediment texture and color. Three long boreholes including one at the seismite site (Fig. 3b) and two within the paleochannel (Fig. 2a) were obtained by tube-well drilling. Sediment samples from the 43-, 61-, and 91-m-long boreholes were described in 2–3 m intervals. An overview of all boreholes and exposures is given in Supplementary Data 1. The extent and character of sand dikes and other soft-sediment deformation and seismite features were surveyed on cleaned exposed walls and floors of a freshly dug and unfilled pond, and their orientation was mapped using a theodolite. Bulk major and trace element concentrations, including strontium, were measured on 20 g of dried bulk sediment using a portable XRF (Thermo-Scientific Niton XL3 Analyzer). The grain-size distributions of each luminescence sample and of sediment obtained from the BNG-MS3 borehole were

measured by laser diffraction using a Malvern Mastersizer 3000 particle-size analyzer.

### Chronology
Fourteen samples for OSL dating (Table S2.2) were obtained with a light-tight vertical lined sampler which collects 30-cm of undisturbed sediment within a borehole and by hammering light-tight, lined PVC tubes horizontally into exposed pond walls. All samples were analyzed for grain-size distribution and bulk strontium content (Table S2.2) which is an indicator of provenance of sand-rich deposits[34]. Luminescence dating was performed using the 4–11 or 4–20 micron silt fraction[29] (Table S2.3) which was isolated and measured using procedures of ref. 107 and described below. A multiple-signal single-aliquot regenerative dose (MS-SAR) protocol[108] was used to screen the samples for sensitivity and paleodose. Two standard SAR protocols with recuperation, recycling, and IR depletion tests were applied to purified quartz; one with a 240 °C preheat and four regenerative doses for high-paleodose (>5 Gy) samples and one with a 200 °C preheat and three regenerative doses for low-paleodose (<5 Gy) samples. Dose recovery ratios of $1.013 \pm 0.006$ ($n = 41$) and $1.036 \pm 0.004$ ($n = 45$), respectively, verified the measurement sequences (Fig. S7). Early background subtraction[109] was used to optimize the relative contribution of the fast component to the quartz OSL signal. The paleodose was calculated as a mean and standard error of at least 3 and most often 6 multigrain 10-mm diameter (~2 mg) silt aliquots per sample. Dose rate was obtained using mainly standard procedures including measurement of radionuclide activities of dried, ashed, and ground bulk sediment on a high-resolution broad-range gamma spectrometer. A water content correction based on grain size and seasonality was applied to unsaturated samples to estimate their time-averaged water content (Fig. S4). Luminescence ages were determined as the paleodose divided by the dose rate and are reported with one sigma uncertainties. Notably, one sample yielded no measurable quartz and thus no luminescence age. The last occurrence of active sand transport in the paleochannel was determined as a weighted mean of the two pointbar sand samples (Fig. 2a). Overbank aggradation rate was calculated by differencing the ages and depths for OSL samples MA I-2 and MA I-3 (Fig. 3a), which were located near the top of the floodplain unit (below the surface mixing zone) and near its base, respectively. Detailed luminescence-dating methodology and interpretation is given in the Supplementary Information and resulting data for each sample are provided in Supplementary Data 3.

### Event magnitude and liquefaction risk
The Supplementary Information provides a detailed consideration of the tectonic setting of the Bengal basin and of prior Bengal Basin earthquakes on record. This information was used to support our reconstruction of the paleo-earthquake including likely origin and magnitudes that are possible for the Bengal Basin. The magnitude of the event was estimated based on general relationships with distance to origin and dike width that were established by prior studies using global datasets of known-magnitude events[37,39]. Dredge-fill construction sites were quantified using Google Earth Engine[110] to establish the location, timing, and area of site development for five-year intervals over the last two decades (2003–2023).

## Data availability
The data generated in this study are provided in the Supplementary Information and Supplementary Data files. Supplementary Data 1–3 are offered as once excel file with three sheets.

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

## Acknowledgements

The research was funded by a U.S. National Science Foundation postdoctoral fellowship (NSF EAR-1855264) and a National Center for Earth Surface Dynamics postdoctoral fellowship (NSF EAR-1246761), both to E.L.C., a Graduate School for Production Ecology & Resource Conservation (PE & RC) of Wageningen University visiting scientist grant to S.L.G. and E.L.C., and a U.S. National Science Foundation grant (NSF EAR-1714892) to M.S.S. and L.S.

## Author contributions

E.L.C., S.L.G., T.R., M.S.S. and J.W. designed the original study, and S.H.A., G.M., A.A.N., A.R. and M.R. supported in-country study design and local research contexts. E.L.C., R.B., S.L.G., G.M., A.A.N., A.R. and M.R. performed the field data collection. E.L.C., J.W. and T.R. conducted luminescence dating, and E.L.C. and S.L.G. conducted strontium and grain-size analyses. E.L.C., R.B., S.L.G., C.V.H., L.S., M.S.S., T.R. and J.W. developed and analyzed the dataset. E.L.C. and S.L.G. drafted the manuscript with contributions from C.V.H., L.S., M.S.S., T.R. and J.W. All authors reviewed and commented on the manuscript.

## Competing interests

The authors declare no competing interests.
