## [Peer Review File · Nature Communications]

Cascading hazards of a major Bengal basin earthquake and abrupt avulsion of the Ganges RiverREVIEWER COMMENTS

Reviewer #1 (Remarks to the Author):

Authors present a well-supported link between seismicity and river avulsion within deltaic lowlands. They argue, convincingly in my opinion, that such a large-scale dramatic response to seismicity should be taken into account for tectonically-affected deltaic environments within a cascade of hazards. The study was executed at the highest standards in the field and the writing is clear and to the point.

I do not support the claim that seismic hazards have not been recognized previously in deltas. In fact, the twin of their GB delta (i.e. the Indus) is a classical for this topic. This does not diminish the importance of their study, which is better supported. However, I suggest the authors to review work on the Allah Bund fault and the blockage of the Nara course of the Indus in the Indus delta by authors like C. P. Rajendran and K. Rajendran as well as Martitia Tuttle. Such work should be cited and integrated as precursors into their paper.

Otherwise, I am looking forward to see this study in print.

Reviewer #2 (Remarks to the Author):

Overall, this paper represents a thorough, well-documented study showing new evidence for the abrupt avulsion of the Ganges River leading to the abandonment and relocation of the mainstem channel as a consequence of a major paleo-earthquake. The paper is well illustrated with some exceptional documentation of field data.

The study represents a good example of a modern work as the authors use a combination of high-quality field data supported by absolute dating. I therefore commend the authors for their excellent, thorough and systematic documentation of data and their holistic approach.

This is a study that presents a useful and significant contribution to science, stressing the high risk associated with the cascading effects of earthquakes in deltas and other lowland settings around the world. The scope and aims of the study are of interest to a wide range of scientists, particularly those interested in tectonically active settings, paleo-earthquakes & seismites or hazard management.

The authors generally do a good job outlining the rationale and aims of the project. They use suitable methods to address the aims and present high-quality field data coupled with luminescence dating. The conclusions are largely supported by the data. The manuscript is fairly well written although there is certainly scope to better organize the observations in order to make it more concise, and better support the interpretations. The manuscript is therefore suitable for Nature Communication and would be of interest to a range of scientists in different fields.

Improvements

-Please provide a general background of sand intrusions, their trigger mechanism and why they are used

as palaeo-earthquake indicators.

-Please provide more details of dikes and fracture orientation in relationship to the paleochannel (a stereonet could be useful) to better support the exclusion of riverbank slumping as a trigger.

-Ensure that you are separating observations and interpretations into different sections. A sub-section is needed to discuss all evidence that supports the earthquake origin of the fluidization and intrusions.

-Please provide a more comprehensive background of the link between the width of the dikes and the magnitude of the earthquakes and show your interpretations in light of current knowledge, limitations and uncertainties regarding that.

Other minor suggestions

-The sentences are very long in places, consider breaking them into two or three separate sentences.

-The presentation of field data is excellent, however, you should make sure that the descriptions are as organized and signposted as possible.

See detailed annotated comments on the attached PDF.

Reviewer #3 (Remarks to the Author):

Review of the manuscript

Cascading hazards of a major Bengal basin earthquake and abrupt 1 avulsion of the Ganges River

by Elizabeth Chamberlain and co-authors.

This study is a very nice piece of work attempting to link very peculiar sedimentary and depositional structures, related to a river avulsion event, to a past major, remote EQ. The careful collection of various excellent field observations in combination with OSL dating of various types of normal, sheeted, and intrusive deposits provides a convincing picture of the 'cascading' impacts the stratigraphic and sedimentary system had experienced in response to the actual local river avulsion process. The evidence is nicely supported by the figures, supplementary material, and describes a scientifically spectacular story.

What needs clarification is to what extent these findings are unique observations so far, or whether comparable structures have previously been found in other geographic areas of the delta. See Lines 228 to 230 in this context: Does the comparison with impacts of the Assam EQ on the banks of the Brahmaputra River mean that comparable liquefaction features are already known from other parts of the delta? If so, it needs to be described in detail how the sedimentary elements look like there, how they compare to the features found in this study, and what makes the new discovery unique. This effort would support the desire to publish this work in *Nature*.

The second issue appears to be related to their conclusions that a major EQ of a magnitude exceeding M 7.5 is required to trigger such a remote geomorphological response. Particularly around Lines 156, 191, and 195 in the manuscript, the authors build on speculations that would greatly benefit from a more geotechnical corroborating approach: If an EQ leads to liquefaction of certain sediment beds that are composed of relatively sorted coarse-silty/fine-sandy material, I wonder to what extent it might be appropriate to assume that those original beds were under-consolidated as well as young. I would assume 'under-consolidation' is a relative description in this context? Commonly, an under-consolidation of non-cohesive material should be relatively age-independent. Thus, I suggest – here and elsewhere across the manuscript text – that the authors attempt to add some professional geotechnical calculations about the fluidization potential of the deposits that participated in the liquefaction event. It would support their suggestion that the observed liquefaction features can technically be related to an EQ magnitude of 7.5 to 8.0. How this specific EQ estimate is currently presented is not very convincing. In Line 258, the authors refer to 'saturated, un-consolidated (what is the difference between un- and under-consolidated then?) Holocene sands'. It is thus, neither clear which consolidation degree the original deposits needed to have, nor how old the deposits might have been.

Such a geotechnical theoretical calculation would also help to exclude other conditions that might lead to a major river avulsion. In Lines 252 to 254, the authors refer to a seemingly similar 'cascading' situation in the Brahmaputra River system. They might want to describe the sedimentological/depositional expression of the major shift in river avulsion there. Are there any features known that might be comparable to the elements observed in this study? Could it be that a flood-related major avulsion process can lead to sufficient enough overloading and pore-pressure changes that lead to the formation of similar geological structures? If even a strong monsoonal event may trigger a major avulsion event (s. a.), more substantial information needs to be provided to assume a magnitude of M 7.5+.

Given the expertise of some of their co-authors, it would be helpful to inform the reader if there has been any EQ reported in geological archives closer to the subduction zone or the Himalayans that dates around 2.5 ka? Is there any seismo-stratigraphic evidence for an EQ that happened 2.5 ka ago? How about the liquefaction beds found within the offshore stratigraphic record?

Why might modern engineering river bank management, seeking to stabilize the river course, negatively impact the natural 'cascading' process that might happen during a catastrophic EQ-triggered avulsion event? The Implications chapter currently sounds like there is no solution to be prepared or to offer protection in such a major avulsion case, may it be related to monsoonal flooding or a remote EQ. Would the authors agree with this conclusion? If so, the last chapter on anthropogenic and technical modification is not very needed; if not, what would be their advice?

The following comments are of minor nature but might help to improve the manuscript.

Line 41 What does 'vary between fan and coastal settings' mean? The apron near the mountains, the delta plain, a deep-sea fan? Please provide some term definition/explanation for readers from the outside.

Line 42 What are 'channel belts'? S. a.

Line 177 What are 'under-filled channel scars'? S. a.

Line 123 Provide sedimentological information, specifically on the degree of sorting and the conclusions that can be drawn from it. How about skewness as a source or transport sorting indicator?

Lines 211 to 217 read somehow repetitive since they do not provide new information or context.

Line 197 What location/region does 'occurring here' refer to? Are the observations of various, supposedly EQ-related geological/sedimentological features a unique finding, or are there other examples in the Ganges river delta? The provided citation refers to an example in the US. What is the link between the two systems?

Author response to reviewer comments

We thank the three reviewers for their thoughtful comments, which have improved the manuscript. Please find our response to each comment below in blue.

Reviewer #1 (Remarks to the Author):

Authors present a well-supported link between seismicity and river avulsion within deltaic lowlands. They argue, convincingly in my opinion, that such a large-scale dramatic response to seismicity should be taken into account for tectonically-affected deltaic environments within a cascade of hazards. The study was executed at the highest standards in the field and the writing is clear and to the point.

I do not support the claim that seismic hazards have not been recognized previously in deltas. In fact, the twin of their GB delta (i.e. the Indus) is a classical for this topic. This does not diminish the importance of their study, which is better supported. However, I suggest the authors to review work on the Allah Bund fault and the blockage of the Nara course of the Indus in the Indus delta by authors like C. P. Rajendran and K. Rajendran as well as Martitia Tuttle. Such work should be cited and integrated as precursors into their paper.

Otherwise, I am looking forward to see this study in print.

It was very encouraging to read this; thank you. Thanks also for pointing us toward the interesting case of the Allah Bund fault. We agree this research is relevant, and we have added text and references to include it as well as referencing Tuttle's contributions on the New Madrid seismites.

To clarify our claim, we changed the last sentence of the Implications to "...may face unrecognized hazards as the result of seismic risk". We do not intend to claim that seismic risk alone to deltas is not recognized; there is ample literature on seismic risk to tectonic basins including Bangladesh, and we cite this literature where relevant. Rather, we demonstrate that the cascade of hazards resulting from earthquakes – specifically the avulsion of a major river channel distal to the earthquake origin and coupled with extensive liquefaction – is not previously well considered for deltas.

Reviewer #2 (Remarks to the Author):

Overall, this paper represents a thorough, well-documented study showing new evidence for the abrupt avulsion of the Ganges River leading to the abandonment and relocation of the mainstem channel as a consequence of a major paleo-earthquake. The paper is well illustrated with some exceptional documentation of field data.

The study represents a good example of a modern work as the authors use a combination of high-quality field data supported by absolute dating. I therefore commend the authors for their excellent, thorough and systematic documentation of data and their holistic approach. This is a study that presents a useful and significant contribution to science, stressing the high risk

associated with the cascading effects of earthquakes in deltas and other lowland settings around the world. The scope and aims of the study are of interest to a wide range of scientists, particularly those interested in tectonically active settings, paleo-earthquakes & seismites or hazard management.

The authors generally do a good job outlining the rationale and aims of the project. They use suitable methods to address the aims and present high-quality field data coupled with luminescence dating. The conclusions are largely supported by the data. The manuscript is fairly well written although there is certainly scope to better organize the observations in order to make it more concise, and better support the interpretations. The manuscript is therefore suitable for Nature Communication and would be of interest to a range of scientists in different fields.

Thank you; we are happy to read this positive overview!

Improvements

-Please provide a general background of sand intrusions, their trigger mechanism and why they are used as palaeo-earthquake indicators.

Good idea. We have added this to the first paragraph of the Paleo-earthquake Archive – Sand Dikes section.

-Please provide more details of dikes and fracture orientation in relationship to the paleochannel (a stereonet could be useful) to better support the exclusion of riverbank slumping as a trigger.

Good idea; we now include a stereonet in the supplement (Fig. S9).

-Ensure that you are separating observations and interpretations into different sections. A subsection is needed to discuss all evidence that supports the earthquake origin of the fluidization and intrusions.

Separating data and discussions into different sections is not the typical format of *Nature Communications*; rather we wrote the manuscript with topical sections addressing the different aspects of our study, including both data and interpretation in each topical section. As the first reviewer commended the writing and clarity, we opt not to significantly restructure the manuscript.

-Please provide a more comprehensive background of the link between the width of the dikes and the magnitude of the earthquakes and show your interpretations in light of current knowledge, limitations and uncertainties regarding that.

We already do this to the extent that is possible for this paleo-earthquake reconstruction, and we have carefully edited our phrasing to better make clear the uncertainties.

Other minor suggestions

-The sentences are very long in places, consider breaking them into two or three separate sentences.

Thank you for the careful proofreading. We have reviewed each comment on the annotated pdf and made changes where we felt it improved the manuscript (e.g., clarifying that these are not lithified sediments, referring to other seismic features as “subordinate” rather than “secondary”, moving a figure from the supplement (now Fig. 4) to better detail sediment lithology).

-The presentation of field data is excellent, however, you should make sure that the descriptions are as organized and signposted as possible.

Agreed, and thank you for the comments on figures and data presentation (e.g., labelling of Fig. 1 panel c as d). We have incorporated these in our revised manuscript.

See detailed annotated comments on the attached PDF.

Thanks; we have reviewed these comments and incorporated your feedback.

Reviewer #3 (Remarks to the Author):

This study is a very nice piece of work attempting to link very peculiar sedimentary and depositional structures, related to a river avulsion event, to a past major, remote EQ. The careful collection of various excellent field observations in combination with OSL dating of various types of normal, sheeted, and intrusive deposits provides a convincing picture of the ‘cascading’ impacts the stratigraphic and sedimentary system had experienced in response to the actual local river avulsion process. The evidence is nicely supported by the figures, supplementary material, and describes a scientifically spectacular story.

Thank you for these positive and encouraging comments.

What needs clarification is to what extent these findings are unique observations so far, or whether comparable structures have previously been found in other geographic areas of the delta. See Lines 228 to 230 in this context: Does the comparison with impacts of the Assam EQ on the banks of the Brahmaputra River mean that comparable liquefaction features are already known from other parts of the delta? If so, it needs to be described in detail how the sedimentary elements look like there, how they compare to the features found in this study, and what makes the new discovery unique. This effort would support the desire to publish this work in *Nature*.

Archives such as these are exceptionally rare for reasons we outline in the second paragraph of the manuscript. To be candid, as a team that has worked 30+ years in the Bengal basin, we were astounded when we discovered these dikes and even more so when we realized the connection to the avulsed paleochannel. To clarify the novelty, we added “Such an archive is previously undocumented in the Bengal basin” to the third paragraph of the manuscript.

The absence of published prior Bengal basin examples of exposed liquefaction features and their structures supports the rarity of the archives we detail. Liquefaction features associated with the 1897 earthquake were previously mapped in research we cite, however, these maps recorded only the surface expression (sand blows and fissures) and not the vertical profiles. As such, there are no documented liquefaction features in the Bengal basin that are comparable

with the ones we describe, to the best of our knowledge. We added a sentence to the Paleo-earthquake Archive – Sand Dikes section to make this clear.

The second issue appears to be related to their conclusions that a major EQ of a magnitude exceeding M 7.5 is required to trigger such a remote geomorphological response. Particularly around Lines 156, 191, and 195 in the manuscript, the authors build on speculations that would greatly benefit from a more geotechnical corroborating approach: If an EQ leads to liquefaction of certain sediment beds that are composed of relatively sorted coarsesilty/fine-sandy material, I wonder to what extent it might be appropriate to assume that those original beds were under-consolidated as well as young. I would assume ‘under-consolidation’ is a relative description in this context? Commonly, an under-consolidation of non-cohesive material should be relatively age-independent. Thus, I suggest – here and elsewhere across the manuscript text – that the authors attempt to add some professional geotechnical calculations about the fluidization potential of the deposits that participated in the liquefaction event. It would support their suggestion that the observed liquefaction features can technically be related to an EQ magnitude of 7.5 to 8.0. How this specific EQ estimate is currently presented is not very convincing. In Line 258, the authors refer to ‘saturated, un-consolidated (what is the difference between un- and under-consolidated then?) Holocene sands’. It is thus, neither clear which consolidation degree the original deposits needed to have, nor how old the deposits might have been.

We carefully reviewed and edited our phrasing to make clear the uncertainties of the paleo-earthquake reconstruction. “Required” is a strong word from the reviewer here, and we do not make the claim that an M 7.5 earthquake or greater **is required** for the event we describe. Rather, we provide a realistic reconstruction of the paleo-earthquake magnitude based on global databases of the relationship of dike width and distance to origin. The approach we use for paleo-earthquake magnitude reconstruction is widely accepted. We acknowledge its limitations and uncertainties, for example by stating “both size and distance relationships **suggest** that the ~2.5 ka earthquake was **likely in the range of M 7.0-8.0**”. Finally, the combination of both dike width and the site’s great distance from the seismogenic centers supports a large magnitude event. Through this careful framing we attempt to avoid such misunderstandings as occurred here and to be open and realistic about the uncertainty of paleo-reconstructions.

While placing constraints on the magnitude of the paleo-earthquake is of interest, its specific magnitude is of secondary importance to the study. Primary here is that, regardless of magnitude, the paleo-earthquake we document generated significant and previously unrecognized cascading hazards including abrupt rerouting of a a very large river channel, and we state this in the manuscript.

Regarding quantitative properties of the sediments, we now provide in the main text grain-size distributions of typical sand and mud deposits for the upper 30m of the Bengal basin and for samples taken at the seismite location, with their associated liquefaction potential (now Fig. 4).

Such a geotechnical theoretical calculation would also help to exclude other conditions that might lead to a major river avulsion. In Lines 252 to 254, the authors refer to a seemingly similar ‘cascading’ situation in the Brahmaputra River system. They might want to describe the sedimentological/depositional expression of the major shift in river avulsion there. Are there any features known that might be comparable to the elements observed in this study?

Could it be that a flood-related major avulsion process can lead to sufficient enough overloading and pore-pressure changes that lead to the formation of similar geological structures? If even a strong monsoonal event may trigger a major avulsion event (s. a.), more substantial information needs to be provided to assume a magnitude of M 7.5+.

In the interest of minimizing speculation, we do not include geotechnical theoretical calculations for reconstruction of this paleo-event. We did consider making geotechnical estimates when first preparing the manuscript, and we re-evaluated in light of this comment. At both times, we judged it ineffective due to the large assumptions required regarding principal stresses, pore water pressure, overburden, etc. As previously stated, we do not “assume” an M7.5+ earthquake; rather we adopt a data-supported approach to magnitude reconstruction drawn from two different empirical parameters as described in our response to the prior paragraph.

Given the expertise of some of their co-authors, it would be helpful to inform the reader if there has been any EQ reported in geological archives closer to the subduction zone or the Himalayans that dates around 2.5 ka? Is there any seismo-stratigraphic evidence for an EQ that happened 2.5 ka ago? How about the liquefaction beds found within the offshore stratigraphic record?

To the best of our knowledge there are no previously published studies that report a Himalaya or Bengal basin earthquake circa 2.5 ka. In fact, chronology in general is lacking for the Holocene deposits/archives of the basin (see Chamberlain et al., 2020 ESPL for a review on this topic), and this is something our team aims to develop.

Why might modern engineering river bank management, seeking to stabilize the river course, negatively impact the natural ‘cascading’ process that might happen during a catastrophic EQ-triggered avulsion event? The Implications chapter currently sounds like there is no solution to be prepared or to offer protection in such a major avulsion case, may it be related to monsoonal flooding or a remote EQ. Would the authors agree with this conclusion? If so, the last chapter on anthropogenic and technical modification is not very needed; if not, what would be their advice? The following comments are of minor nature but might help to improve the manuscript.

In general, embanked rivers have fewer natural outlets for water and are therefore particularly vulnerable to bank erosion, enhanced stream power, and thereby avulsion during high-stage events. This is explained in the first two paragraphs of the Implications. Regarding flood risk, we believe that the best hazard mitigation policy is one underpinned by scientific knowledge, and our study develops this knowledge.

We agree that the implications of our findings can be better illustrated. Toward this, we adapted content from the supplement (formerly Figs S4, S10a,c, and S11a) to the main text to illustrate the character and extent of modern dredge sand platforms (now Fig. 7) and their vulnerability to liquefaction based on sediment texture (now Fig. 4), and we updated the title to include sediment liquefaction. Fig. 4 also satisfies the recommendations of Reviewers 2 and 3 to provide more information in the main text describing sediment texture. Offering a specific management approach for Bangladesh to earthquake-triggered cascading hazards is beyond the scope of this manuscript and should involve a multi-disciplinary team including Bangladeshi policy makers and social scientists as well as physical scientists.

Line 41 What does ‘vary between fan and coastal settings’ mean? The apron near the mountains, the delta plain, a deep-sea fan? Please provide some term definition/explanation for readers from the outside.

We improved to “between relatively high-gradient, coarse-sediment fan settings and relatively low-gradient, fine-grain coastal settings”, so we now mention specific aspects of these geomorphic zones that may influence avulsion.

Line 42 What are ‘channel belts’? S. a.

This is common terminology in (fluvial) geomorphology and we do not feel it serves a short-format manuscript to define it here.

Line 177 What are ‘under-filled channel scars’? S. a.

We describe this in several sentences in the preceding section on the Paleochannel archive, paragraph 2, and we added reference to two figures which further illustrate the terminology.

Line 123 Provide sedimentological information, specifically on the degree of sorting and the conclusions that can be drawn from it. How about skewness as a source or transport sorting indicator?

We moved the grain-size distribution figure (now Fig. 4) to the main manuscript. Additional information is provided in the supplementary files (e.g., Table S2.2). Further analysis of Bengal basin sediments can be found in prior, referenced work, most recently Raff et al. (2023, *Nature Communications*).

Lines 211 to 217 read somehow repetitive since they do not provide new information or context.

We prefer to keep this important synthesis statement for clarity and to avoid misunderstanding by non-specialist readers.

Line 197 What location/region does ‘occurring here’ refer to? Are the observations of various, supposedly EQ-related geological/sedimentological features a unique finding, or are there other examples in the Ganges river delta? The provided citation refers to an example in the US. What is the link between the two systems?

We replaced “occurring here” with “we document” to make it clear that this interpretation is about the Bengal basin sand dikes we show in this study. The reference is for a broader dataset relating earthquake magnitude to distance to origin that is presented in Fig. 6 and discussed in the preceding lines. To clarify we added a reference to Fig. 6a in parentheses after this sentence.